suicide; risk factors; global mental health; global public health; epidemiology

**Corresponding author:**
Cristian Cabello Jiménez;
Email: cristian.cabello123@gmail.com

# Suicide reattempts in a Chilean border region: A retrospective case–control study of sociodemographic, psychosocial and clinical risk factors

Cristian Cabello Jiménez[1] , Sergio Muñoz Navarro[2], Rodrigo Casanueva[1], Cristina Mamani Cabrera[1] and Franco Mascayano[1,3]

[1]Instituto de Salud Pública, Universidad Andrés Bello, Chile; [2]Departamento de Salud Pública, Universidad de La Frontera, Chile and [3]Department of Mental Health, Johns Hopkins Bloomberg School of Public Health, USA

## Abstract

Suicide mortality is rising in Latin America, yet evidence on repeat suicidal behavior remains scarce. Chile's national surveillance system offers a unique opportunity to study high-risk populations. This study identifies sociodemographic, psychosocial, and clinical factors associated with suicide reattempts in Arica–Parinacota, a Chilean border region with high suicide rates and significant indigenous and migrant populations. We conducted a retrospective case–control study using regional surveillance data (2017–2021). We compared individuals with multiple attempts (cases) against those with a single attempt (controls). Multivariable logistic regression was used to estimate adjusted odds ratios (AOR) for reattempt, controlling for seven covariates. Among 299 individuals, six factors independently predicted reattempt: female gender (AOR = 2.27, 95% CI: 1.26–4.22), unemployment (AOR = 2.96, 95% CI: 1.31–6.87), family history of suicide (AOR = 2.59, 95% CI: 1.31–5.28), psychiatric disorder (AOR = 3.92, 95% CI: 2.26–6.92), domestic violence (AOR = 2.71, 95% CI: 1.53–4.87), and psychoactive substance use (AOR = 2.07, 95% CI: 1.13–3.86). Findings delineate a risk profile characterized by intergenerational suicide, gendered vulnerability, psychiatric comorbidity, and social stressors. This study demonstrates the public-health value of disaggregated, surveillance-based analytics for improving triage and targeted screening in middle-income settings.

## Impact statement

This study identifies key risk factors for suicide reattempts in Arica and Parinacota, Chile's northernmost region. Using five years of data from one of Latin America's few event-level suicide attempt surveillance systems, we conducted a retrospective case–control analysis to better understand who is most at risk of repeating suicidal behavior.

Our analysis identified six independent risk factors for suicide reattempt: female sex, unemployment, psychiatric disorder, family history of suicide, exposure to domestic violence and use of psychoactive substances. These findings point to clear and actionable risk profiles that can support early detection, guided follow-up care and inform targeted prevention strategies in both clinical and community settings.

This is the first study to examine suicide reattempts in Chile's northern border region, a socially diverse area with significant indigenous and migrant populations. By generating context-specific evidence, it contributes directly to strengthening the country's National Suicide Prevention Program – particularly its post-attempt care provision. The findings highlight the urgent need to integrate mental health screening and follow-up protocols into routine primary care services and to prioritize risk factors such as psychiatric symptoms, substance use, violence and unemployment in clinical assessments.

Finally, the study underscores the importance of moving beyond event-based reporting. Establishing a standardized and mandatory national registry for suicide attempts would enable long-term follow-up, improve surveillance and generate actionable data to support more equitable and context-sensitive prevention strategies.

## Introduction

Suicide mortality is rising in many regions, such as Latin America (Dávila-Cervantes, 2022), positioning suicidal behavior as a major global public health concern. Each year, over 720,000 people die by suicide worldwide, with many more attempting it (WHO, 2018).

Suicide attempts – and especially reattempts – are critical areas for prevention, as a previous attempt is the strongest predictor of future suicide death (Espandian et al., 2020). It is estimated that one in five people who attempt suicide will reattempt (de la Torre-Luque et al., 2023), contributing significantly to the burden on healthcare systems and highlighting the limitations of current prevention, detection and treatment efforts (López-Goñi et al., 2020; Vuagnat et al., 2020; Kim et al., 2024).

In response, the World Health Organization (2018) has urged countries to develop national plans to reduce suicide mortality. However, many countries, including those in Latin America, still lack comprehensive national suicide prevention strategies. Chile is among the few exceptions, having declared suicide a national public health priority in 2011 and implementing the National Suicide Prevention Program in 2013 (de Salud, 2013; UNICEF, 2017). This comprehensive program has been deployed across all 16 regions through the *Secretaría Regional Ministerial de Salud* (Regional Health Ministry Office), and a key component is the gradual implementation of a national, event-level suicide attempt surveillance system that collects sociodemographic, psychosocial and clinical data at the regional level (de Salud, 2013). This system offers a unique opportunity to study suicide reattempts in populations typically underrepresented in routine health statistics.

As part of this effort, in 2017, the Arica and Parinacota region became one of the first in the country to implement an integrated notification system for suicide attempts, collecting clinical and contextual information such as education level, employment status, nationality, indigenous status and history of domestic violence.

Arica and Parinacota, located in the northernmost part of Chile and bordering Peru and Bolivia, have a predominantly urban population, with only a small share residing in rural areas (INE, 2018). The region is characterized by a high presence of migrant population and indigenous people (INE, 2017; 2018). Socioeconomically, 9.2% of its inhabitants live in income poverty, while 18.6% of the population experiences multidimensional poverty, reflecting deprivations in education, health, employment and housing conditions beyond income alone (CASEN, 2023).

With respect to suicide, the region has presented one of the highest adolescent suicide rates in the country, reaching 16.8 per 100,000 inhabitants in 2017, placing it second nationwide (de Salud, 2019). In addition, although its general suicide mortality rate in 2016 was slightly below the national average, the region experienced the second-highest annual increase between 2015 and 2016, rising by 4.0 points (Gobierno de Chile, 2016). Taken together, these factors position Arica and Parinacota as a region of heightened structural vulnerability, reinforcing the relevance of studying the factors associated with suicide reattempt in this specific context. While these characteristics underscore the relevance of studying suicide behavior in this region, it is important to acknowledge that existing surveillance data may not provide sufficient power to analyze these groups in detail due to small representation. In addition, structural and social barriers, such as stigma, socioeconomic disadvantages and barriers to accessing health services, may be one of the disadvantages of this surveillance system.

Global studies have identified several risk factors for reattempts, including prior attempts, depression, impulsivity, family history of suicide, traumatic experiences such as physical or sexual abuse and substance abuse (Ahmed et al., 2020; Liu et al., 2022; Morales Téllez et al., 2023; Pemau et al., 2024; Escobedo-Aedo et al., 2025). However, discrepancies persist in the literature, and more research is needed in underrepresented contexts such as Latin America (de la Torre-Luque et al., 2023; Pemau et al., 2024), where mental health care systems are less robust and prevention strategies remain underdeveloped (Mascayano et al., 2018).

In Chile, research has focused primarily on suicide mortality, partly due to the high quality of mortality records (Sandoval and Portaccio, 2022). Although suicide attempts have received more attention in recent years, studies on suicide reattempts remain scarce. Most research has concentrated on adolescents and has relied on samples from specific cities or towns, often using small or geographically limited samples, which makes it difficult to generalize findings to broader regional contexts (Silva et al., 2017). Given Chile's vast cultural and socioeconomic diversity across its regions, regional-level evidence is crucial to better understand how local factors influence suicidal behavior.

By examining a region marked by high diversity and disadvantage, this study contributes to the generation of context-sensitive evidence that can inform more effective and equitable suicide prevention strategies. Accordingly, this exploratory study aimed to identify risk factors associated with suicide reattempts in a sample of individuals with confirmed suicide attempts reported in the regional surveillance system of Arica and Parinacota between 2017 and 2021.

## Methods

### Study design and data collection

We conducted a retrospective case–control study using health records of suicide attempts reported in the Arica and Parinacota region of Chile between 2017 and 2021. The regional surveillance system relies on a paper-based notification form that healthcare providers complete at the time of patient care. Reporting suicide attempts is voluntary for healthcare institutions, as no national or regional mandate requires clinicians to submit notifications. Once completed, forms are digitized and transmitted electronically to the Regional Health Ministry Office (*Secretaría Regional Ministerial de Salud).*

The standardized notification form captures sociodemographic (age, sex, educational level, indigenous status, nationality and occupation), psychosocial (history of domestic violence, family history of suicide and psychoactive substance use) and clinical information (hospital referral, physical illness, method of suicide attempt, presence of psychiatric illness and enrollment in a mental health treatment program) based solely on patient self-report. Because the surveillance system is passive and depends on voluntary submission, coverage is incomplete and reporting practices vary across institutions and over time.

### Study sample

Between 2017 and 2021, the regional surveillance system registered 566 suicide attempt notifications. Of these, 267 records were excluded: 227 due to insufficient minimum data, 2 because they corresponded to suicide deaths, and 38 because they were duplicate records of multiple attempts by the same individual. In cases with multiple notifications, only the most recent attempt was retained to avoid duplication.

Records classified as having "insufficient minimum data" were those in which key exposure variables were missing, including psychosocial factors such as psychiatric disorder, domestic violence, psychoactive substance use and family history of suicide, as well as essential sociodemographic variables. Because these variables were required for the main analyses, cases lacking this

information could not be included. A comparison of included and excluded cases showed no substantial differences in sociodemographic or psychosocial characteristics based on the information available.

In this study, a suicide attempt was defined as a self-inflicted injury with the explicit intention of causing death, consistent with international and national guidelines (de Salud, 2013). The index attempt corresponds to the event documented in the regional notification form during the study period. Information on suicide attempts prior to the index event was obtained exclusively through patient self-report, as no centralized registry exists to verify previous attempts. A suicide reattempt was defined as the occurrence of an additional suicide attempt in an individual with a history of at least one previous attempt.

The primary outcome was the presence of a previous suicide attempt. Cases were defined as individuals who self-reported at least one lifetime suicide attempt prior to the index event. Controls were those who reported no previous attempts, making the index event their first known attempt.

### Variables

We examined sociodemographic, psychosocial and clinical exposure variables recorded in the standardized notification form. Sociodemographic variables included sex (binary), age, educational attainment (low, medium and high), occupational status (unemployed, student and paid employment), indigenous identity and nationality. Age was analyzed descriptively as a continuous variable and then categorized into <25 years and ≥ 25 years to distinguish developmental stages relevant to suicidal behavior.

Indigenous identity and nationality were originally collected as multicategory variables, but several categories had very small cell counts. For analytical purposes, both variables were collapsed into binary categories (Indigenous/Non-Indigenous; Chilean/Foreign national). Because records with missing values in key sociodemographic or psychosocial variables had already been excluded during sample selection, these variables did not contain missing data in the analytical dataset.

Psychosocial variables included exposure to domestic violence, family history of suicide and psychoactive substance use, all coded as binary (yes/no). In this context, psychoactive substance use reflects a behavioral report rather than a formal clinical diagnosis.

Clinical variables were also analyzed in binary form and included referral to hospitalization, physical illness, psychiatric disorder and current enrollment in a mental health treatment program.

The variable "method of suicide attempt" was excluded from the analyses due to a high proportion of missing data and inconsistencies in cases where multiple methods were recorded, which made it impossible to assign a single valid category for analysis.

### Statistical analysis

We first conducted a descriptive analysis of all variables. Age was summarized using mean and standard deviation, and categorical variables were described using absolute frequencies. We also performed a stratified descriptive analysis by age group (<25 and ≥ 25 years) to explore potential differences in exposure patterns.

Bivariate associations between each exposure variable and the outcome (previous suicide attempt) were examined using simple binary logistic regression models to obtain unadjusted odds ratios (ORs) and 95% confidence intervals.

To identify factors associated with suicide reattempt, we fitted a multiple logistic regression model. Covariate selection was informed by clinical relevance and prior research. Variables were included when they represented established predictors or potential confounders in the relationship between psychosocial and clinical exposures and suicide reattempt. We first specified a model (Model 1) including age, sex, family history of suicide and a set of exposure variables of substantive interest: occupation, psychiatric disorder, mental health treatment, domestic violence and psychoactive substance use.

During model evaluation, we detected collinearity between psychiatric disorder and current enrollment in a mental health treatment program, as most individuals receiving treatment also reported a psychiatric diagnosis. This association reflects that treatment enrollment does not function as an independent exposure in this context; rather, it acts as a clinical indicator of severity or a proxy for the formal recognition of the patient's condition. Including both variables would have resulted in overadjustment, potentially masking the true impact of the patient's clinical history. Indeed, in Model 1, the treatment variable lost statistical significance due to its strong association with psychiatric disorder.

To address this, we specified a second model (Model 2). To ensure this model effectively captures the foundational clinical vulnerability of the participants, we prioritized psychiatric disorder and excluded the treatment variable. This approach provides a more robust estimation of the underlying psychopathological burden that characterizes individuals with a history of attempts, independent of their current level of professional support or system engagement. Consequently, Model 2 retained psychiatric disorder, alongside with age, sex, family history of suicide, domestic violence and psychoactive substance use.

Model 2 stability was confirmed using variance inflation factors (VIFs). Following methodological recommendations suggesting that, in epidemiological and behavioral research, collinearity between conceptually related variables can become meaningful at lower VIF values than conventional cutoffs, we adopted a conservative threshold of 2.5 (Johnston et al., 2018). All variables in Model 2 had VIF values below this threshold.

Data analyses were conducted using R version 4.4.2 R (R Core Team, 2025) and RStudio version 2023.12.0 + 467. Descriptive analyses were performed using the `gtsummary` package (Sjoberg et al., 2021), and a multiple logistic regression model was fitted with the `glm` function. Tables were generated combining `tbl_summary` and `tbl_uvregression` outputs, and exported to Word using `gt` (Iannone et al., 2025) and `officer` (Gohel and Heckmann, 2025) packages.

## Results

Table 1 shows descriptive findings from the sample of 299 individuals with a recorded suicide attempt. The mean age was 25 years

**Table 1.** Sociodemographic, psychosocial and clinical characteristics of the suicide attempt sample (*n* = 299)[a]

| Variable | *N* (%)/mean ± SD[b] |
| --- | --- |
| Age (years) | 25.98 ± 13.69 |
| Sex | |
| Men | 101 (34%) |
| Women | 198 (66%) |

*(Continued)*

**Table 1.** (Continued)

| Variable | N (%)/mean ± SD[b] |
|---|---|
| **Indigenous group** | |
| Alacalufe | 1 (0.3%) |
| Aymara | 38 (13%) |
| Diaguita | 1 (0.3%) |
| Mapuche | 9 (3.0%) |
| None | 237 (79%) |
| Other | 11 (3.7%) |
| Quechua | 1 (0.3%) |
| Rapa Nui | 1 (0.3%) |
| **Nationality** | |
| Bolivian | 4 (1.3%) |
| Chilean | 289 (97%) |
| Colombian | 1 (0.3%) |
| Peruvian | 4 (1.3%) |
| USA | 1 (0.3%) |
| **Education** | |
| Low | 50 (17%) |
| Medium | 177 (59%) |
| High | 72 (24%) |
| **Occupation** | |
| Employed | 108 (36%) |
| Student | 145 (48%) |
| Unemployed | 46 (15%) |
| **Referred to hospitalization** | |
| Yes | 247 (83%) |
| **Previous suicide attempt** | |
| Yes | 142 (47%) |
| **Family history of suicide** | |
| Yes | 56 (19%) |
| **Physical illness** | |
| Yes | 24 (8.0%) |
| **Psychiatric disorder** | |
| Yes | 109 (36%) |
| **Mental health treatment** | |
| Yes | 96 (32%) |
| **Domestic violence** | |
| Yes | 72 (24%) |
| **Psychoactive substance use** | |
| Yes | 97 (32%) |

[a]Several indigenous groups and nationalities had very small counts, making robust statistical analysis for these groups impossible. Therefore, these variables were collapsed into binary categories for subsequent analyses.
[b]Mean ± SD; *n* (%).

(SD = 13.6; range: 7–81 years). A total of 36.4% had a diagnosed psychiatric disorder, with depression being the most common (54.1%), followed by personality disorders (14.6%), schizophrenia (4.5%) and bipolar disorder (3.6%). Other conditions reported at lower frequencies (<2%) included substance use disorder, attention deficit hyperactivity disorder, obsessive compulsive disorder and anxiety disorders. In 17.4% of cases, the specific psychiatric diagnosis was not reported.

Regarding psychoactive substance use, the most frequently reported substance was marijuana (49.4%), followed by polysubstance use (25.7%), alcohol (6.3%), cocaine (4.1%), medications such as clonazepam and diazepam (4.1%), and cocaine base paste (3%). In 6.1% of cases, the type of substance was not specified.

When stratified by age, participants under 25 years reported slightly higher prevalence of psychoactive substance use (35.1% *vs.* 27.8%), exposure to domestic violence (25.1% *vs.* 22.2%) and having a family history of suicide (20.4% *vs.* 15.7%) compared to older participants, while psychiatric disorders (29.3% *vs.* 49.1%) and being enrolled in a mental health treatment program (31.9% *vs.* 32.4%) were less common in this group.

Table 2 presents the bivariate analysis to identify variables associated with suicide reattempts and to estimate unadjusted ORs. Being unemployed (OR = 2.68, 95% CI: 1.33–5.55), a family history of suicide (OR = 2.33, 95% CI: 1.29–4.31), a psychiatric disorder (OR = 4.33, 95% CI: 2.63–7.25), being enrolled in a mental health treatment program (OR = 3.00, 95% CI: 1.82–5.02), experience of domestic violence (OR = 2.41, 95% CI: 1.41–4.22) and psychoactive substance use (OR = 2.70, 95% CI: 1.65–4.49) were all significantly associated with higher odds of suicide reattempts.

Table 3 presents the multivariate logistic regression analysis, estimating adjusted odds ratios (AORs) for suicide reattempts. As previously noted, receiving mental health treatment was excluded from the final model due to collinearity with psychiatric disorder.

Being female (AOR = 2.27, 95% CI: 1.26–4.22), unemployed (AOR = 2.96, 95% CI: 1.31–6.87), having a family history of suicide (AOR = 2.59, 95% CI: 1.31–5.28), having a psychiatric disorder (AOR = 3.92, 95% CI: 2.26–6.92), experiencing domestic violence (AOR = 2.71, 95% CI: 1.53–4.87) and using psychoactive substance (AOR = 2.07, 95% CI: 1.13–3.86) were all significantly associated with increased odds of suicide reattempts, after controlling for age and sex. No imputation was conducted; only completed cases were included in the final analysis.

## Discussion

### *Main findings*

This study examined risk factors associated with suicide reattempt in a sample of individuals with suicide attempts recorded in the northernmost region of Chile. We found that female gender, being unemployed, having a family history of suicide, psychiatric disorder, experiences of domestic violence and the use of psychoactive substances increased the risk of suicide reattempts after controlling for age and sex.

### *Regional context*

The profile observed in our study – predominantly young women under 25 years old, many of whom are students with psychiatric diagnoses – is consistent with findings from several Latin American and Asian studies (Vélez et al., 2022; Morales Téllez et al., 2023; Kim et al., 2024). However, contrasting patterns have been reported in some contexts in Europe, where suicide reattempts are more common among unemployed or divorced men with psychiatric or interpersonal difficulties (Burón et al., 2016). The only similarity

**Table 2.** Bivariate analysis: risk factors associated with suicide reattempt (*n* = 299)

| Variables | Frequencies and percentages | | OR (95% CI) | | | |
| --- | --- | --- | --- | --- | --- | --- |
| | Controls, *N* = 157[a] | Cases, *N* = 142[a] | *N* | OR | 95% CI | *p*-value |
| Age | | | 299 | | | |
| >24 years | 56 (36%) | 52 (37%) | | – | – | |
| ≤24 years | 101 (64%) | 90 (63%) | | 0.96 | 0.60, 1.54 | 0.9 |
| Sex | | | 299 | | | |
| Man | 60 (38%) | 41 (29%) | | – | – | |
| Women | 97 (62%) | 101 (71%) | | 1.52 | 0.94, 2.49 | 0.089 |
| Indigenous group | | | 299 | | | |
| No | 120 (76%) | 117 (82%) | | – | – | |
| Yes | 37 (24%) | 25 (18%) | | 0.69 | 0.39, 1.22 | 0.2 |
| Nationality | | | 299 | | | |
| No | 149 (95%) | 140 (99%) | | – | – | |
| Yes | 8 (5.1%) | 2 (1.4%) | | 0.27 | 0.04, 1.08 | 0.10 |
| Education | | | 299 | | | |
| Low | 27 (17%) | 23 (16%) | | – | – | |
| Medium | 95 (61%) | 82 (58%) | | 1.01 | 0.54, 1.91 | >0.9 |
| High | 35 (22%) | 37 (26%) | | 1.24 | 0.60, 2.57 | 0.6 |
| Occupation | | | 299 | | | |
| Employed | 66 (42%) | 42 (30%) | | – | – | |
| Student | 74 (47%) | 71 (50%) | | 1.51 | 0.91, 2.51 | 0.11 |
| Unemployed | 17 (11%) | 29 (20%) | | 2.68 | 1.33, 5.55 | 0.007 |
| Referred to hospitalization | | | 299 | | | |
| No | 30 (19%) | 22 (15%) | | – | – | |
| Yes | 127 (81%) | 120 (85%) | | 1.29 | 0.71, 2.38 | 0.4 |
| Family history of suicide | | | 299 | | | |
| No | 137 (87%) | 106 (75%) | | – | – | |
| Yes | 20 (13%) | 36 (25%) | | 2.33 | 1.29, 4.31 | 0.006 |
| Physical illness | | | 299 | | | |
| No | 148 (94%) | 127 (89%) | | – | – | |
| Yes | 9 (5.7%) | 15 (11%) | | 1.94 | 0.84, 4.77 | 0.13 |
| Psychiatric disorder | | | 299 | | | |
| No | 124 (79%) | 66 (46%) | | – | – | |
| Yes | 33 (21%) | 76 (54%) | | 4.33 | 2.63, 7.25 | <0.001 |
| Treatment in a mental health program | | | 299 | | | |
| No | 124 (79%) | 79 (56%) | | – | – | |
| Yes | 33 (21%) | 63 (44%) | | 3.00 | 1.82, 5.02 | <0.001 |
| Domestic violence | | | 299 | | | |
| No | 131 (83%) | 96 (68%) | | – | – | |
| Yes | 26 (17%) | 46 (32%) | | 2.41 | 1.41, 4.22 | 0.002 |
| Psychoactive substance use | | | 299 | | | |
| No | 122 (78%) | 80 (56%) | | – | – | |
| Yes | 35 (22%) | 62 (44%) | | 2.70 | 1.65, 4.49 | <0.001 |

[a]*n* (%).
Abbreviations: CI = confidence interval, OR = odds ratio.

**Table 3.** Multiple logistic regression analysis: independent factors associated with increased risk of suicide reattempt ($n = 299$)

| Variables | AOR | 95% CI | p-value |
|---|---|---|---|
| **Sex** | | | |
| Man | – | – | |
| Women | 2.27 | 1.26, 4.22 | 0.008 |
| **Age** | | | |
| > 24 years | – | – | |
| ≤ 24 years | 0.72 | 0.32, 1.59 | 0.4 |
| **Occupation** | | | |
| Employed | – | – | |
| Student | 2.11 | 0.94, 4.93 | 0.075 |
| Unemployed | 2.96 | 1.31, 6.87 | 0.010 |
| **Family history of suicide** | | | |
| No | – | – | |
| Yes | 2.59 | 1.31, 5.28 | 0.007 |
| **Psychiatric disorder** | | | |
| No | – | – | |
| Yes | 3.92 | 2.26, 6.92 | <0.001 |
| **Domestic violence** | | | |
| No | – | – | |
| Yes | 2.71 | 1.53, 4.87 | <0.001 |
| **Psychoactive substance use** | | | |
| No | – | – | |
| Yes | 2.07 | 1.13, 3.86 | 0.020 |

Abbreviations: CI = confidence interval, AOR = adjusted odds ratio.

between both contexts is the role of unemployment as a risk factor; however, the demographic and psychosocial characteristics of the individuals most affected differ substantially.

Notably, in our study, 47.1% of individuals under 25 years old had a suicide reattempt, a proportion that is considerably higher than those reported in studies conducted among young people aged 10–24 in Saudi Arabia (26.1%), 10–18 in England (27.3%) and youths in France (30%) (Hawton et al., 2012; Ahmed et al., 2020; Mirkovic et al., 2020). In our sample, nearly 48% of participants were students, suggesting that school-related contexts may play a relevant role. Although this age group did not present markedly higher rates of psychoactive substance use, domestic violence, or psychiatric disorder in our sample, existing literature emphasizes that adolescence and early adulthood are particularly vulnerable periods due to academic pressure, emotional immaturity, identity conflicts and limited coping strategies (Liu and Wang, 2024). Furthermore, a longitudinal Norwegian study found that psychological distress in adolescence, including self-harm, was strongly associated with suicidal thoughts and behaviors continuing into early adulthood (ages 22–25) (Sivertsen et al., 2024), highlighting the persistence of vulnerability beyond adolescence.

In the regional context of Arica and Parinacota, these developmental vulnerabilities intersect with broader structural factors, such as economic insecurity, high internal and international migration, residential mobility and unequal educational

trajectories, that may exacerbate stress during adolescence and emerging adulthood. Evidence from Latin American settings shows that socioeconomic disadvantage during youth is associated with higher levels of psychological distress and suicidal behavior (Benjet et al., 2018). Moreover, studies highlight that young migrants may experience elevated risks of depression, anxiety and self-harm due to migration-related stressors, family disruption and social exclusion (McCord et al., 2019). Although empirical research in Latin America is limited, international studies consistently show that indigenous adolescents face increased vulnerability to suicidal behaviors due to discrimination, cultural marginalization and barriers to accessing appropriate mental health care (Hatcher, 2016; De Zilva et al., 2022). These contextual pressures may help explain why the reattempt rate among young people in our sample is substantially higher than that reported in other countries.

### Risk factors for suicide reattempt

Regarding associated factors, international literature has shown that female sex is often associated with higher rates of suicide reattempts, where women are overrepresented in the sample (de la Torre-Luque et al., 2023) – consistent with our findings, where 66% of cases were women. Gendered patterns in emotional expression and help-seeking behavior may influence both the likelihood of suicide reattempt and their detection by healthcare systems. Women are more likely than men to disclose emotional distress and to seek mental health care, facilitating earlier identification of suicide risk (Shi et al., 2021; Güney et al., 2024). However, paradoxically, some research suggests that women may delay seeking specialized psychiatric care longer than men after a suicide attempt, which may influence their clinical trajectories (Juárez-Domínguez et al., 2024). These patterns underscore the need for further research to better understand how gender differences shape both clinical trajectories and outcomes in suicide behavior.

Psychiatric disorder was the strongest predictor in our analysis, with depression and personality disorders being the most frequent diagnoses, which aligns with numerous previous studies across different contexts (Vélez et al., 2022; Mehanović et al., 2024; Pemau et al., 2024). Given the consistency of this association in international literature, psychiatric disorders represent a critical target for suicide prevention strategies. However, recent studies also suggest that a non-negligible proportion of individuals attempt suicide without a prior psychiatric diagnosis (Oquendo et al., 2024). This challenges the assumption that suicide risk is confined to clinical populations and underscores the need to broaden prevention efforts beyond psychiatric settings.

Other significant factors associated with suicide reattempt included having a family history of suicide, experiencing domestic violence, psychoactive substance use and being unemployed. These findings are consistent with previous research that links family suicide history to both genetic predisposition and shared environmental stressors (Qin et al., 2003; Morales Téllez et al., 2023).

Exposure to domestic violence, especially during childhood and adolescence, has been shown to have long-term psychological effects, increasing vulnerability to self-harming behaviors (Devries et al., 2013; Xiao et al., 2024). This is consistent with Chilean studies showing that women who have experienced partner or family violence are nearly three times more likely to attempt suicide (Inostroza et al., 2022), highlighting the enduring mental health consequences of interpersonal violence. Psychoactive substance

use, particularly cannabis and polydrug consumption, was common among suicide reattempts and remains a public health concern in Chile, where adolescent cannabis use is among the highest in Latin America (OEA, 2019).

Finally, although there are no robust estimates quantifying the individual-level impact of unemployment on suicide reattempt, our findings align with population-level evidence. Recent meta-analyses report significantly higher odds ratios for suicide mortality, suicide attempt and suicide ideation among unemployed individuals (Amiri, 2022). In the Chilean context, economic downturns like the 2008 subprime recession, which increased unemployment, were followed by rises in suicide mortality, underscoring the structural link between economic instability and suicidality (Baeza et al., 2022). These findings highlight the need to understand suicide reattempts as a multifactorial phenomenon and to design programs that integrate clinical, social and structural determinants using an intersectional perspective.

### Structural barriers and underrepresented groups

Despite the sociocultural diversity of Arica and Parinacota – characterized by a significant presence of migrant and indigenous populations (INE, 2017; 2018) – our analysis did not yield statistically significant associations between these backgrounds and a history of suicide reattempts. This lack of significance, however, should not be interpreted as an absence of risk, but rather as a reflection of the methodological constraints inherent in the sample size.

This limited representation of these specific subgroups necessitated the collapsing of diverse ethnic and national identities into broad binary categories, a process that potentially masks within-group heterogeneities and dilutes specific risk signals. These results could be interpreted as an indication of unmeasured structural and cultural barriers, such as unequal access to healthcare, compounded by administrative, linguistic and cultural challenges, as well as stigma and discrimination (Garza and Abascal Miguel, 2025; Rocha-Jiménez et al., 2025), that may impede these populations from being captured by the emergency-based surveillance system.

Future research with targeted sampling strategies may be needed to more accurately assess these dynamics.

### Study limitations

The central limitations of this study are linked to the nature of the regional surveillance system, whose voluntary and incomplete reporting determines who is captured in the dataset and who remains unrecorded.

The system relies on voluntary reporting by emergency departments, resulting in incomplete and uneven coverage across the region. In the absence of a mandatory national registry, case capture is limited to events that are reported by healthcare professionals on duty in emergency services. Although all services use the same standardized notification form, data quality may still be compromised by incomplete or missing responses due to reliance on patient self-report. The surveillance system also tends to capture more severe suicide attempts that require emergency medical attention, whereas less severe or self-managed attempts are less likely to be documented. Moreover, access to emergency services is shaped by structural and social factors that influence whether individuals are able or willing to seek care. For instance, migrant populations and Indigenous communities may face legal uncertainties, discrimination, linguistic barriers, geographic isolation or fear of stigma, all of which can reduce their likelihood of presenting to emergency departments and therefore being recorded in the system.

These characteristics imply that the likelihood of being included in the dataset depends on a combination of clinical and contextual factors. Individuals who require emergency medical intervention, who live closer to reporting institutions or who have fewer structural barriers are more likely to be captured, whereas those with less severe attempts or limited access to emergency care are systematically underrepresented. When the likelihood of being recorded is related both to the exposures of interest and to the probability of a future attempt, this differential capture can introduce selection bias. As a result, the associations observed in our study reflect the characteristics of individuals who access emergency services rather than the full population of people who attempt suicide.

### Implications for prevention and policy

Despite these limitations, this study provides valuable insights into the profiles associated with a history of suicide attempts in a context that has been scarcely explored. By focusing on the region of Arica and Parinacota, the study helps fill a significant gap in Chilean and Latin American literature. The use of a case–control design enabled the identification of higher risk profiles at the point of clinical presentation, which is essential for developing locally grounded screening strategies. The study draws on a wide range of socio-demographic, psychosocial and clinical variables – such as sex, employment status, psychiatric history, psychoactive substance use and family background – supporting a nuanced and multifactorial understanding of suicidal behavior. Furthermore, this study underscores the relevance of advancing toward mandatory reporting of suicidal behavior in Chile. By demonstrating that a public health surveillance instrument can effectively identify individuals with higher clinical complexity at presentation, the research highlights its potential utility for frontline health professionals and contributes to ongoing efforts to strengthen early detection and targeted intervention strategies within the health system.

### Conclusion

Based on our findings, we emphasize the need to implement robust screening and triage strategies at the time of the index suicide attempt, particularly among adolescents and young adults, who represent the majority of suicide reattempts in our sample. In the Chilean context, this entails strengthening the implementation of the National Suicide Prevention Program, especially its provisions for post-attempt clinical assessment and immediate referral pathway. Our results highlight the clinical importance of systematically evaluating self-reported history of prior suicide attempts, psychiatric symptoms, exposure to violence, psychoactive substance use and employment status. These factors should be viewed as markers of accumulated psychiatric vulnerability rather than causal predictors, and their identification should be prioritized in comprehensive clinical assessments at the point of care.

From a social and service perspective, the association with unemployment highlights the need for intersectoral policies that address not only mental health but also socioeconomic vulnerabilities. Finally, to improve care continuity, Chile must move beyond event-centered reporting and develop integrated follow-up registries that track individuals over time. Establishing a standardized and mandatory national suicide attempt registry would enhance

epidemiological surveillance and provide actionable data to guide equitable and context-sensitive prevention strategies.

**Open peer review.** To view the open peer review materials for this article, please visit http://doi.org/10.1017/gmh.2026.10193.

**Data availability statement.** The data used in this study are not publicly available online but can be requested from the Secretaría Regional Ministerial de Salud de Arica–Parinacota (Regional Health Ministry Office), by following the official data access procedures of the institution.

**Acknowledgements.** We thank the Secretaría Regional Ministerial de Salud de Arica y Parinacota (Regional Health Ministry Office) and the Department of Epidemiology of Arica for their support in providing the requested data in the appropriate format and within the timeframe required to carry out this research.

**Author contribution.** C.C.: conceptualization; methodology; formal analysis; writing – original draft. S.M.: formal analysis; supervision. R.C.: conceptualization; writing – review and editing. C.M.: data curation; formal analysis. F.M.: conceptualization; supervision; writing – review and editing. All authors reviewed the results and approved the final version of the manuscript.

**Financial support.** The authors declare that this study was conducted without financial support.

**Competing interests.** The authors declare no conflict of interest.

**Ethics statement.** This study was approved by the Review Committee of the Institute of Public Health at Universidad Andrés Bello, which granted an ethics exemption due to the retrospective design and the use of publicly accessible data. Data were obtained in accordance with Chilean legislation on access to public information and were provided by the Regional Health Ministry Office (Secretaría Regional Ministerial de Salud) of Arica–Parinacota. All data were anonymized and handled confidentially, in line with national regulations on statistical confidentiality and data protection.

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
