## [Reviewer Report]

The study provides valuable insights into suicide reattempts in a scantly explored region.

- Table 1 provides detailed frequencies for various indigenous groups and nationalities. However, in subsequent analyses, these are collapsed into binary “Indigenous group: Yes/No” and “Nationality: Yes/No”. The initial detailed breakdown makes it clear that many specific indigenous groups (e.g., Alacalufe, Diaguita, Quechua, Rapa Nui) and nationalities (e.g., Colombian, USA) have extremely small counts (N=1 or N=4). While collapsing is statistically necessary for small numbers, explicitly pointing out the inherent impossibility of robust analysis for these specific, very small subgroups earlier in the methods or results (e.g., in a footnote to Table 1 or when discussing variable recoding) would enhance transparency.

- A significant drawback is that the data relies on health records completed voluntarily by healthcare providers, as reporting suicide attempts is not mandatory. This introduces a strong potential for underreporting and especially selection bias, as less severe or unreported cases may be excluded from the sample, affecting its completeness and representativeness. The authors include this as a limitation of the study, but I suggest a more in depth analysis of how this shortcoming affects their results.

- A substantial number of records (227 out of 566 initial attempts, almost 40% of the initial data) were excluded due to “insufficient minimum data”. This indicates significant data quality issues within the regional surveillance system, which could introduce bias and further compromise the representativeness of the analyzed sample. The exclusion of potentially informative variables like “method of suicide attempt” due to a high proportion of missing data further limits the scope of the analysis.

- The categorization of age into “under 25 years and 25 years or older” is a broad division. While justifiable for certain analyses, the rationale for this specific cutoff could be more thoroughly explained, particularly given the strong emphasis on “adolescent suicide rates” in the introduction and “adolescents and young adults” in the conclusion. A finer age stratification might have captured more nuanced patterns within these critical age groups.

- The introduction effectively establishes the context and the significance of studying suicide reattempts in Arica and Parinacota, especially regarding its diverse indigenous and migrant populations and their exposure to socioeconomic disadvantage and barriers to care. However, this strong emphasis creates an expectation that the study will provide specific insights into these groups. The subsequent finding in the discussion that these groups were not significantly associated due to small sample size and limited power creates a discrepancy between the problem framing and the actual findings for these specific populations. To manage reader expectations, the introduction could be refined to acknowledge the challenges of studying these underrepresented groups with existing surveillance data, thereby setting the stage for the later discussion of limitations more coherently.

---

## [Reviewer Report]

September 09,2025

Dear authors of the manuscript: Suicide Reattempts in a Chilean Border Region: A Retrospective Case-Control Study of Sociodemographic, Psychosocial, and Clinical Risk Factors.

Feedback

1.- The term “Global South” refers to a very broad geopolitical area, so in the context of the manuscript it is ambiguous, considered that it should be eliminated throughout the document since it refers to a very specific area of Chile, in addition in the discussion of the document it is no longer mentioned.

2.- Please add the geographical location of Arica and Parinacota, first describe the social and economic characteristics.

3.- In the background section, highlight the need for study on the indigenous and migrant population in the context of a border region and not only because of the scarcity of previous studies.

4.- Lines 162-166 should be in the study sample section.

5.- Define the variables suicide attempt and suicide reattempt.

6.- On line 223 it says that substance use disorder was rare, but 32% had SUD, please check it.

7.- Please justify why you did not include in the regression analysis in depressive disorder, highly related to suicidal behavior, the same as personality disorders.

8.- In line 270 it says, the results contrast with what was reported in Europe, but they coincide, in their study unemployment is associated with suicide retry, it is suggested to modify the wording.

9.- In line 278 it mentions that the group under 25 years of age did not have a marked rate of substance use, domestic violence or psychiatric disorders, but that data is not in the results, please write it down.

10.- Some references are very old, for example, Núñez F 2006, Méndez-Bustos 2013, Tedstone 2010, please update them or reconsider including them.

11.- It is suggested to include the following references:

Escobedo-Aedo, P. J., Méndez, P., Álvarez, R., Baca-García, E., & Porras-Segovia, A. (2025). Predictors of Suicide Attempts and Reattempts in a Sample of Chilean Adolescents. Early intervention in psychiatry, 19(3), e70024. https://doi.org/10.1111/eip.70024

Romero-Pimentel AL, Mendoza-Morales RC, Fresan A, García-Dolores F, González-Sáenz EE, Morales-Marín ME, Nicolini H and Borges G (2018) Demographic and Clinical Characteristics of Completed Suicides in Mexico City 2014–2015. Front. Psychiatry 9:402. DOI: 10.3389/FPSYT.2018.00402

---

## [Editor Report]

In addition to the recommendations by reviewers, please consider an explanation of the rationale for selecting covariates in multivariable models that does not rely on statistical significance, which is not a recommended method for variable selection. Instead, I would recommend that multivariable models are built based on prior causal knowledge via building a direct acyclic graph and selecting variables accordingly.

---

## [Reviewer Report]

November 24, 2025

Authors: Suicide Reattempts in a Chilean Border Region: A Retrospective Case-Control Study of Sociodemographic, Psychosocial, and Clinical Risk Factors.

Dear authors,

Thank you for listening to the comments, new comment.

Line 352 says “... in our study 47.1% of individuals under...” It should say “... in our study 47.1% of individuals had a ...”, please verify.

Table 1. For yes/no variables, record only the value corresponding to yes.

---

## [Editor Report]

Thank you. Please revise the suggestion by reviewers. Below see my own comments, please address them too:

1. Clarify the rationale and limitations of the surveillance system.

While the manuscript emphasizes the novelty of the regional reporting system, it would benefit from clearer articulation of its limitations. Specifically: voluntary reporting, incomplete coverage, and differential data quality may bias estimates of prevalence and risk factors. The sentence about “structural and social barriers… which may be one of the disadvantages of this surveillance system” should be revised for clarity—the link between social disadvantage and notification gaps needs clearer framing.

2. Improve consistency and avoid redundant text in Methods.

The Methods section repeats key definitions—particularly the definition of the primary outcome and case/control status—almost verbatim. These duplications should be removed to avoid confusion. In addition, the description of variable recoding (e.g., collapsing indigenous identity and nationality into binary forms) should explicitly clarify how missing data were handled.

3. Strengthen reporting on missing data and exclusion criteria.

The study excluded 227 cases for “insufficient minimum data.” The manuscript should briefly describe what minimally required data elements were missing and whether excluded cases differed systematically from included ones. This transparency is important given the potential for selection bias, especially in a voluntary notification system.

4. Clarify statistical methodology and model rationale.

While the multicollinearity issue between psychiatric disorder and mental health treatment is noted, the explanation of model selection is somewhat difficult to follow. Simplifying the description of Model 1 vs Model 2, and briefly explaining why treatment loss of significance indicates collinearity (rather than confounding or mediation), would improve interpretability. Additionally, the text references VIF thresholds; consider briefly noting the threshold used and why it was chosen.

5. Improve the Discussion by integrating findings with regional implications.

The Discussion is strong but could better connect the main findings—particularly the high reattempt rate among young people—to structural vulnerabilities described earlier in the Introduction. Expanding briefly on how socioeconomic disadvantage, migration, and indigenous identity intersect with the identified risk factors would enhance contextual depth.

---

## [Reviewer Report]

Third review

08 January 2026

Dear Manuscript Authors: Suicide Reattempts in a Chilean Border Region: A Retrospective Case-Control Study of Sociodemographic, Psychosocial, and Clinical Risk Factors.

Thank you for addressing the previous comments.

Please. Add the following citations to the reference list:

Line Citation

887 CASEN, 2023

1036 Core Team

1103 Burón, 2016

1165 Qin, 2003

1169 Devries, 2013; Xiao, 2024

1175 OAS, 2019

References that are not in the text:

Line 1337: Encuesta de caracterización socioeconómica

Line 1457: Sjoberg....

Please correct the numbering of the tables.

---

## [Editor Report]

Thanks for addressing reviewers' concerns. In addition, please see my own new comments below:

The manuscript would benefit from tighter alignment between the analytic design, model selection, and the scope of its recommendations. First, conclusions and policy implications should be reframed to reflect that the outcome is a cross-sectional, self-reported history of prior attempts, identifying higher-risk profiles at presentation rather than causal predictors of future reattempts. Second, the rationale for the final multivariable model should be strengthened by explicitly justifying the exclusion of mental health treatment beyond collinearity, acknowledging its potential role as a mediator or severity proxy and clarifying that the retained model primarily captures underlying psychiatric vulnerability. Finally, interpretations regarding migrants, Indigenous populations, and youth should more clearly reflect limited statistical power and variable collapsing, framing null findings as inconclusive if appropriate and subgroup-specific implications as hypothesis-generating rather than definitive.